# A Modified Method to Assess Tidal Recruitment by Electrical Impedance Tomography

**DOI:** 10.3390/jcm8081161

**Published:** 2019-08-03

**Authors:** Thomas Muders, Benjamin Hentze, Philipp Simon, Felix Girrbach, Michael R.G. Doebler, Steffen Leonhardt, Hermann Wrigge, Christian Putensen

**Affiliations:** 1Department of Anesthesiology and Intensive Care Medicine, University Hospital Bonn, Bonn 53127, Germany; 2Chair for Medical Information Technology, RWTH Aachen University, Aachen 52074, Germany; 3Department of Anesthesiology and Intensive Care Medicine, University of Leipzig, Leipzig 04103, Germany; 4Department of Anesthesiology, Intensive Care and Emergency Medicine, Pain Therapy, Bergmannstrost Hospital Halle, Halle 06112, Germany

**Keywords:** acute respiratory distress syndrome, positive end-expiratory pressure, electrical impedance tomography, computed tomography, monitoring, functional imaging

## Abstract

Avoiding tidal recruitment and collapse during mechanical ventilation should reduce the risk of lung injury. Electrical impedance tomography (EIT) enables detection of tidal recruitment by measuring regional ventilation delay inhomogeneity (RVDI) during a slow inflation breath with a tidal volume (V_T_) of 12 mL/kg body weight (BW). Clinical applicability might be limited by such high V_T_s resulting in high end-inspiratory pressures (P_EI_) during positive end-expiratory pressure (PEEP) titration. We hypothesized that RVDI can be obtained with acceptable accuracy from reduced slow inflation V_T_s. In seven ventilated pigs with experimental lung injury, tidal recruitment was quantified by computed tomography at PEEP levels changed stepwise between 0 and 25 cmH_2_O. RVDI was measured by EIT during slow inflation V_T_s of 12, 9, 7.5, and 6 mL/kg BW. Linear correlation of tidal recruitment and RVDI was excellent for V_T_s of 12 (*R*^2^ = 0.83, *p* < 0.001) and 9 mL/kg BW (*R*^2^ = 0.83, *p* < 0.001) but decreased for V_T_s of 7.5 (*R*^2^ = 0.76, *p* < 0.001) and 6 mL/kg BW (*R*^2^ = 0.71, *p* < 0.001). With any reduction in slow inflation V_T_, P_EI_ decreased at all PEEP levels. Receiver-Operator-Characteristic curve analyses revealed that RVDI-thresholds to predict distinct amounts of tidal recruitment differ when obtained from different slow inflation V_T_s. In conclusion, tidal recruitment can sufficiently be monitored by EIT-based RVDI-calculation with a slow inflation of 9 mL/kg BW.

## 1. Introduction

In patients with acute respiratory distress syndrome (ARDS), inspiratory overdistension and cyclic opening and collapse of atelectatic but recruitable lung units during tidal ventilation (tidal recruitment) contribute to ventilator-induced lung injury (VILI) [1,2,3,4]. The current standard of care for ARDS limiting tidal volumes (V_T_) and end-inspiratory pressures (P_EI_) has been shown to reduce VILI and improve mortality in ARDS [5]. Debate exists about the positive end-expiratory pressure (PEEP) level that minimizes tidal recruitment and induces comparatively less VILI [6,7]. Meta-analyses [8,9] did not demonstrate a reduction in mortality using high PEEP strategies in general. This has been partially explained by individual differences in the potential for alveolar recruitment in response to higher PEEP [10]. Clinical studies suggest that in ARDS patients with collapsed but potentially recruitable lung tissue, individualized PEEP titration may be beneficial to detect and improve lung recruitment [6] and to minimize tidal recruitment [7].

A conclusion on regional ventilation distribution and regional tidal recruitment [11] cannot be drawn from global indices of lung function. In contrast, electrical impedance tomography (EIT), a non-invasive imaging technique, allows bedside monitoring of regional ventilation [12,13]. We have recently introduced and have validated an EIT-based index; the regional ventilation delay inhomogeneity (RVDI) [14,15] that quantifies temporal heterogeneity in regional ventilation time courses during a slow inflation V_T_ maneuver of 12 mL/kg body weight (BW). RVDI allows the estimation of PEEP-associated changes of tidal recruitment [15].

When applying our EIT-based method using RVDI to assess tidal recruitment during PEEP-trials in ARDS patients, use of repeated slow inflations V_T_ maneuvers of 12 mL/kg BW may result in potentially harmful high end-inspiratory overdistension, especially at high PEEP levels, which might limit its clinical applicability. RVDI measurements based on reduced slow inflation V_T_ maneuvers are required to decrease P_EI_.

We hypothesized that tidal recruitment can also be assessed using RVDI when derived from slow inflation maneuvers with reduced V_T_. This hypothesis was investigated by an extended secondary analysis of experimental data derived from a porcine lung injury model.

## 2. Animals and Methods

### 2.1. Animals and Ethics

This study presents an extended analysis of data from a previous investigation in a porcine lung injury model [15]. Experiments were approved by the local ethics committee. Animal housing, care, and experiments were performed in adherence with the Guide for the Care and Use of Laboratory Animals (National Academy of Science 1996). We followed the 3Rs-principle (Appendix B). This study is being reported according to the Animal Research Reporting of In Vivo Experiments (ARRIVE) guidelines (Appendix A).

### 2.2. Study Protocol

#### Anesthesia, Ventilatory Settings, and Induction of Lung Injury

Seven healthy pigs (29–34 kg) were anesthetized, tracheotomized and instrumented in the supine position as previously described [14,15]. Pigs were mechanically ventilated (Engström Carestation, GE Healthcare, Helsinki, Finland) in a volume-controlled mode with a V_T_ of 6–8 mL/kg BW, a respiratory rate (RR) of 15 min^−1^, an inspiratory: expiratory ratio of 1:1, a fraction of inspired oxygen (F_i_O_2_) = 0.5 and a PEEP = 5 cmH_2_O. Depth of anesthesia was verified by paw pinch before animals were paralyzed, and the absence of spontaneous breathing activity was confirmed by observation of continuously displayed gas-flow tracing [14,15]. Following preparation, lung injury was induced by i.v. injection of oleic acid (0.1 mL/kg) in combination with intraabdominal hypertension of 20 cmH_2_O caused by intraperitoneal infusion of saline [14,15]. RR was increased up to 30 min^−1^ to avoid hypercapnia (P_a_CO_2_ > 50 mmHg) and F_i_O_2_ was titrated to maintain a P_a_O_2_ above 80 mmHg. Intrinsic positive end-expiratory pressure was excluded by observation of end-expiratory zero flow patterns at the ventilator. Animals were continuously paralyzed in order to suppress spontaneous breathing efforts.

After lung injury induction and stabilization, animals were transferred to a computed tomography (CT) scanner. PEEP levels of 0, 5, 10, 15, 20 and 25 cmH_2_O were applied in a randomized order (sealed envelopes), while all other ventilatory settings remained unchanged. To avoid effects of lung volume history, lungs were derecruited (by disconnection of the respirator) and then recruited by a continuous positive airway pressure of 50 cmH_2_O for 40 s ensuring full lung recruitment [14], before changing PEEP settings, respectively. Any maneuver was preceded by an intravenous fluid bolus of 100 mL of saline to prevent severe hemodynamic depression, thus maintaining mean arterial blood pressure above 55 mmHg. After 30 min of ventilation on the selected PEEP EIT measurements were performed during a single slow inflation maneuver with a constant gas flow by setting the respiratory rate to 4 min^−1^, which resulted in an inflation time of 7.5 s and a V_T_ of 12 mL/kg BW. Then, spiral-CT scans of the lungs were performed during end-expiration and end-inspiration holds.

### 2.3. Measurements and Data Analysis

Details on cardiovascular, ventilatory, lung mechanics, and blood gas measurements have been reported previously [15].

#### 2.3.1. Computed Tomography

Spiral-CT scans (120 kV, 120 mA) covering the complete lung tissue were performed during end-expiration and end-inspiration holds using a Brilliance 64 CT scanner (Philips Healthcare, Hamburg, Germany). Tidal volume was 6–8 mL/kg BW. Images were reconstructed in slices of 8 mm using a standard filter. Densitometric analysis of all pulmonary CT slices was performed using a computer program (Osiris, University of Geneva, Switzerland) as described previously [14,15]. The amount of non-aerated lung tissue (defined as densities between −100 and +100 Hounsfield units) was derived from all CT slices. Tidal recruitment was calculated for the total lung as differences between amounts of non-aerated lung tissue at end-in- and end-expiration [14,15].

#### 2.3.2. Electrical Impedance Tomography

An EIT system (EIT evaluation KIT II, Dräger Medical GmbH, Lübeck, Germany) was used. Image reconstructions were performed using a modified Newton-Raphson algorithm and images of ventilation distribution were generated by comparing impedance changes to a reference state [15].

#### 2.3.3. Quantification of Heterogeneity in Ventilatory Time Courses

Regional and global time courses of impedance changes were recorded with a temporal resolution of 20 Hz during a single slow inflation maneuver [15]. The global impedance-time curve *∆Z(t)* was calculated as the sum of the impedance changes of all pixels.

Regional-Ventilation-Delay time (*Δt*_RVD_) [15] was determined between the start of inspiration defined as first increase of the global *∆Z(t)* curve and the time when the respective regional curve *∆Z_i_(t)* reached a threshold of 40% of the maximal local impedance change (Figure 1). To address the fact that *Δt*_RVD_ depends on inflation time (*Δt_max − min_*), *Δt*_RVD_ was normalized by division through *Δt_max-min_*:RVD = *Δt*_RVD_/*Δt_max − min_*.(1)

The regional-ventilation-delay index (RVD) describes the delay given in (%) of inflation time until the respective regional impedance change exceeds a certain threshold [15].

RVDs were obtained pixel-to-pixel and a color-coded map was plotted to visualize the pixels’ RVDs (Figure 1). To quantify temporal heterogeneity, RVD-inhomogeneity (RVDI) was calculated as the standard deviation of all single-pixel RVDs (SD_RVD_, Figure 1) after filtering and masking [15].

Originally, RVDI was shown to correlate well with tidal recruitment, when calculated from a slow inflation maneuver with a V_T_ of 12 mL/kg BW [15]. Under the assumption of constant gas flow, the applied volume will increase linearly with time. Thus, in order to obtain slow inflation maneuvers with reduced tidal volumes of 9, 7.5, and 6 mL/kg BW it is possible to truncate the full EIT data of 12 mL/kg BW after 75%, 62.5%, and 50% of the elapsed time. Our custom MATLAB software is then applied to the truncated EIT data to simulate RVDI analysis at reduced tidal volumes. Details are given in Figure 1.

Accordingly, P_EI_ resulting from reduced slow inflation V_T_ maneuvers were estimated based on the pressure measurements during the original 12 mL/kg BW maneuvers.

#### 2.3.4. Selection of Definite RVDI Thresholds to Predict Tidal Recruitment

The entire data on tidal recruitment were divided into quartiles. The threshold for tidal recruitment separating the 1st from the 2nd through the 4th quartiles was 2% of the total lung volume. The threshold for tidal recruitment separating the 1st and 2nd from the 3rd and 4th quartiles was 4% of the total lung volume (Figure 2).

Receiver-Operator-Characteristic (ROC) curve analyses were performed to estimate sensitivity and specificity of certain RVDI thresholds to predict an increase in tidal recruitment. For this purpose, all data pairs (tidal recruitment in % of lung volume and RVDI in % inflation time) of the CT-validation cohort were grouped in “below” vs. “above” 2% tidal recruitment (threshold separating the 1st from the 2nd through the 4th quartiles, see above), and “below” vs. “above” 4% tidal recruitment (threshold separating the 1st and 2nd from the 3rd and 4th quartiles, see above), respectively. “Below” was classified as a control group for the ROC curve analyses. Assessments were performed for RVDI values obtained from 12, 9, 7.5, and 6 mL/kg BW inflations, respectively.

RVDI thresholds with a fixed sensitivity of 90% were derived from ROC curves. Finally, an increase of RVDI above these thresholds, when decreasing PEEP, predicts an increase in tidal recruitment above 2% and 4% with a sensitivity of 90%, respectively.

### 2.4. Statistical Analysis

Primary outcome measures were RVDI values calculated from different slow inflation volumes. Since no reliable pilot data or data from publications were available for this exploratory study setting, a sample size calculation was not possible. GraphPad Prism (Version 8; GraphPad Software, San Diego, CA, USA) was used for statistical analyses.

RVDI values obtained from different slow inflation tidal volumes at different PEEP levels were compared, using a two-way repeated-measures analysis of variances (ANOVA) after confirming normal distribution (Shapiro-Wilk’s-W-test), linear correlations and Bland and Altman analyses. Changes in P_EI_ resulting from different slow inflation V_T_ maneuvers were compared using a two-way repeated-measures analysis of variances (ANOVA) after confirming normal distribution (Shapiro-Wilk’s-W-test) and differences were separated using post hoc tests (Dukey). Tidal recruitment measured by CT was linearly correlated with RVDI. Linear correlations were calculated between these parameters for single animals (intra-individually) and between these parameters for all animals of the respective cohort (inter-individually) taking data points from all PEEP levels.

Receiver-Operator-Characteristic curve analyses were performed to estimate sensitivity and specificity of certain RVDI thresholds to predict an increase in tidal recruitment.

Data are given as mean ± standard deviation (SD), *p* < 0.05 was considered significant.

## 3. Results

### 3.1. Lung Injury and Cardiopulmonary Effects of Different PEEP Levels

The lung injury induction caused in all animals an oxygenation and lung mechanics impairment [15] compatible with the current criteria for moderate human ARDS [16]. Effects of different PEEP-levels on hemodynamic and pulmonary parameters have previously been described in detail [15]. Briefly, increasing PEEP levels cause a decrease in cardiac output, mean arterial blood pressure, intrathoracic blood volume, and stroke volume. All animals were hemodynamically stable during recruitment maneuvers.

### 3.2 Temporal Heterogeneity Measured by EIT

Heterogeneity of regional ventilatory time courses as quantified by RVDI decreased with higher PEEP-levels (Figure 2A and Figure 3) when derived from the original inflation maneuver using a slow inflation V_T_ of 12 mL/kg BW [15]. Present extended analyses show that this effect could also be observed when RVDI was calculated from reduced slow inflation V_T_ of 6 to 9 mL/kg BW (Figure 2A and Figure 3). Linear correlation between RVDI values calculated from the original slow inflation V_T_ of 12 mL/kg BW maneuver and reduced slow inflation V_T_s are given in Figure 2B and Table 1. The correlation was excellent when comparing slow inflation V_T_s of 12 vs. 9 mL/kg BW and was reduced but still clinically acceptable with further reduction in slow inflation V_T_. Absolute values of RVDI decreased with decreasing slow inflation V_T_ (Figure 2A). Bland and Altman analyses (Table 1) revealed that systematical bias increased with further reduction in slow inflation V_T_.

### 3.3. End-Inspiratory Pressures Resulting from Different Slow Inflation V_T_ Maneuvers

With any reduction in slow inflation V_T_, resulting P_EI_ significantly decreased at all PEEP levels (Figure 4). Decrease in P_EI_ was more pronounced at lower PEEP levels (Figure 4).

### 3.4. Tidal Recruitment at Different PEEP Levels

With an increase in PEEP, the amount of tidal recruitment measured by CT decreased [15]. Half of the measurements showed tidal recruitment of more than 4% of the lung volume (both upper vs. both lower quartiles, Figure 5). For the lowest quartile of the measurements, tidal recruitment was below 2% of the lung volume (1st vs. 2nd quartile, Figure 5).

### 3.5. Correlation of RVDI and Tidal Recruitment Obtained from Different Slow Inflation V_T_ Maneuvers

Linear correlation between tidal recruitment (in % lung volume) quantified by CT and RVDI (in % inflation time) as measured by EIT during the original 12 mL/kg BW slow inflation maneuver was very good to excellent (Figure 5B, Table 2) when calculated inter-individually and intra-individually. Correlations were excellent for RVDI values calculated from a slow inflation V_T_ of 9 mL/kg but slightly decreased with further reduction in slow inflation V_T_ (Figure 5B, Table 2).

### 3.6. Prediction of Tidal Recruitment thresholds from RVDI Values

As derived from quartile analyses on tidal recruitment (Figure 5B), a 2% threshold was chosen to separate the 1st (lowest) from the 2nd through the 4th quartiles of measurements. Accordingly, a 4% threshold was chosen to separate 1st and 2nd from the 3rd and 4th quartiles. ROC-curve analyses were performed to estimate sensitivity and specificity of certain RVDI thresholds to predict an increase in tidal recruitment above the respective threshold (2% or 4% of the lung volume, respectively) with decreasing PEEP levels (Figure 6). Results of the ROC analyses for the different inflation volumes are given in Table 3. The area under the curve was high for all predicting models. Results were slightly better for the 4% threshold (Table 3). When fixing sensitivity at 90% specificities (Table 3) were higher for the 4% tidal recruitment threshold.

## 4. Discussion

The main findings of our study in a recruitable porcine lung injury model are:
RVDI can be derived from slow inflation maneuvers using a low V_T_ of 6 to 9 mL/kg BW, to quantify temporal heterogeneity of regional ventilatory time courses.Absolute RVDI values are not comparable when derived from slow inflation maneuvers using different V_T_s.RVDI values from slow inflation maneuvers using low V_T_s of 6 to 9 mL/kg BW correlate with tidal recruitment.

### 4.1. Effects of Slow Inflation V_T_ on RVDI-Values

Our extended analyses demonstrate that changes in temporal heterogeneity of regional ventilatory time courses can be quantified by RVDI from reduced slow inflation V_T_ maneuvers using EIT. RVDI values calculated from reduced slow inflation V_T_ maneuvers correlate well with RVDI values derived from original slow inflation maneuvers using V_T_ of 12 mL/kg BW. However, slow inflation maneuvers using lower V_T_ result in lower RVDI values. This is plausible because RVDI is the variation of regional delay times, which depends on the total duration of the slow inflation V_T_. As a consequence, bias increases and agreement between RVDI values decrease when slow inflation V_T_ was reduced to 7.5 mL/kg BW or below (Table 1).

As visualized in Figure 1, regional impedance time curves (original regional signal, Figure 1B) show scattered results in the graph. Normalization of impedance changes (*y*-axis) and inflation times (*x*-axis, normalized regional signal, Figure 1B) of the regional curves causes a conversion into a spindle-shaped form. The spread of these normalized curves at the 40% impedance change threshold is quantified by RVDI as visualized by the two-sided arrow in Figure 1B. Reducing V_T_ during slow inflation maneuvers decreased spread of the normalized curves. This suggests that further reduction of V_T_ during slow inflation gradually lowers diagnostic resolution and increases the signal-to-noise ratio of the method. We simulatedreduced slow inflation V_T_s by analyzing the truncated EIT raw data of the original 12 mL/kg BW V_T_. This might not further impair our results, since, under the assumption of a constant and slow gas flow, linear reduction of the regional tidal volume should be uniform.

EIT raw data revealed pronounced artefacts in the cardiac region of single animals that interfered with delay measurements during slow inflation maneuvers using a low V_T_. This might explain the poor intra-individual correlation of RVDI-values when comparing 12 to 6 mL/kg BW in single animals (Table 1, pig 6, right column).

Finally, absolute RVDI values are not directly comparable, when obtained from different slow inflation V_T_ maneuvers. This circumstance prevents comparison of single values of RVDI-measurements derived from different individuals or collectives. Therefore, relative changes in RVDI values might be more significant.

### 4.2. Assessment of Tidal Recruitment Using RVDI Measurements

Linear correlation of RVDI-values and tidal recruitment decreased with a reduction in slow inflation V_T._ However, it was still acceptable even when slow inflation V_T_ was decreased below 9 mL/kg BW. Poor intra-individual correlation of tidal recruitment and RVDI-values derived from 6 mL/kg BW in single animals (Table 2, pig 6, right column) might be explained by cardiac artefacts, as discussed above. Thus, our data suggest that PEEP related changes in tidal recruitment can also be assessed by RVDI-measurements derived from a reduced slow inflation V_T_ of 6–9 mL/kg BW using EIT.

It is important to note that EIT scans were obtained during slow inflation V_T_ maneuvers of 6–12 mL/kg BW, whereas tidal recruitment was measured by CT during subsequent “regular” tidal volume breaths of 6–8 mL/kg BW. As discussed before [15], although obtained during an artificial maneuver with different inflation volume and flow, RVDI-measurements have to be seen as a surrogate describing temporal heterogeneity of regional lung mechanics that, in turn, affects tidal recruitment during tidal breaths.

### 4.3. PEEP Titration Using EIT

#### 4.3.1. EIT-Based Approaches to Individually Titrate PEEP

Several EIT-based approaches have been conducted to individually optimize ventilatory settings [12,13]. Changes in distribution and homogeneity of end-expiratory lung volume [17] and tidal volume [18,19,20,21] were quantified at different airway pressure levels. Zhao et al. [22] demonstrated that the inhomogeneity of spatial ventilation distribution is also influenced by tidal recruitment. Regional lung mechanics were recently derived from EIT to analyze regional collapse [19,23], lung recruitment [24] and overdistension [19,23].

A major advantage of EIT is its high temporal resolution, allowing analysis of aeration and ventilation time courses. Regional filling characters of the lung were initially described in the temporal domain by Hintz et al. [25]. Regional opening pressures have been derived from EIT [26,27]. An “inflation delay” in dependent lung regions during mechanical ventilation has been described by Victorino et al. [28]. Our group introduced and validated the RVD index [14,15] to quantify the temporal heterogeneity of regional ventilation. RVD was also calculated from regular breaths during mechanical ventilation [29], assisted, and spontaneous efforts [30]. However, previous research [15] demonstrated that a slow inflation maneuver is mandatory for sufficient RVDI calculation and assessment of tidal recruitment.

#### 4.3.2. Regional Ventilation Delay Inhomogeneity to Individually Titrate PEEP

Various approaches are conceivable to use EIT-based RVDI measurements to individually titrate PEEP in order to reduce tidal recruitment. We used ROC-curve analyses to determine certain RVDI-thresholds to predict an increase in tidal recruitment above 2% and 4% of the total lung volume. AUCs were acceptable for all predicting models. When aiming at a fixed sensitivity of 90%, specificity was low (40%–60%) for the 2% thresholds but significantly improved (63–74) for thresholds predicting an increase of tidal recruitment above 4%. However, resulting RVDI-thresholds were numerically different when derived from different slow inflation V_T_s. This might limit the clinical applicability of strategies aiming at definite RVDI-thresholds, since separate calibration of RVDI measurements may be necessary for different animal models or groups of patients. However, RVDI-measurements may be used to predicting an increase of tidal recruitment above 4% although threshold may vary with different slow inflation V_T_s.

Independent of slow inflation V_T_ and duration, an increase of temporal heterogeneity in regional ventilatory time courses as indicated by a rise in RVDI indicates an increase in tidal recruitment. Thus, the individual minimum of RVDI might indicate minimal tidal recruitment and allow the selection of the lowest possible PEEP level minimizing tidal recruitment [15]. This promising approach was used by Nestler et al. [31] to individually optimize PEEP during general anesthesia in morbidly obese patients with healthy lungs. Preliminary data of our group [32] show that this strategy might improve regional gas exchange in an animal lung injury model.

### 4.4. Clinical Applicability of RVDI Measurements

As described above an approach minimizing RVDI-values to individually titrate PEEP is, in principle, applicable in humans [31]. However, the safety of the method may be questionable because of the high V_T_s associated with high end-inspiratory airway pressures.

Without a doubt high V_T_s during mechanical ventilation are hazardous. In contrast, slow inflation maneuvers are commonly used in respiratory mechanics with even higher inflation volumes and increased airway pressures [33,34]. The effects of single slow flow breaths with increased V_T_s and airway pressures remain unclear and V_T_ limits, which are valid for mechanical ventilation cannot be fully applied to slow inflation V_T_ maneuvers. However, the present study shows that RVDI-measurements obtained from reduced slow inflation V_T_s of 6–9 mL/kg BW may provide valuable information for PEEP titration just as from a V_T_ of 12 mL/kg. Additionally, any reduction in slow inflation V_T_ significantly decreased P_EI_ at all PEEP levels. With further reduced V_T_ and P_EI_, the method should be less risky for clinical application.

ARDS is characterized by a broad pathophysiological variability, especially with respect to lung collapse, recruitability, and tidal recruitment. Since our results come from simulated data based on an animal study, findings need to be reproduced in ARDS patients with recruitable and non-recruitable lung collapse while deriving RVDI values from several slow inflation V_T_s. Whether PEEP-titration based on measurements of regional ventilatory heterogeneity is clinically advantageous in ARDS patients warrants further investigations.

### 4.5. Limitations

Our study clearly has several limitations. In our animal model of a highly recruitable lung injury, we combined the widely used oleic acid injection model [35] causing endothelial lung injury [36,37,38] with increased IAP [39]. Both are commonly seen in extrapulmonary ARDS, e.g., caused by abdominal sepsis [40]. However, results might not be transferable to patients with ARDS without recruitable lung units, e.g., caused by pneumonia. Further research is warranted to evaluate the reproducibility of our findings in humans.

Furthermore, increased abdominal pressure caused relatively high end-inspiratory pressures, due to a reduction in chest wall compliance. Although it is likely that transpulmonary pressures were lower, we cannot prove so, since esophageal pressures were not obtained.

We used recruitment maneuvers with airway pressures up to 50 cmH_2_O. These pressures were necessary to ensure full lung recruitment in our animal model in order to avoid lung volume history to affect our measurements. However, recruitment maneuvers can be harmful, especially in patients with lower potential for alveolar recruitment. We do not claim these maneuvers as a routine procedure in patients.

For these systematic extended analyses, we simulated reduced slow inflation V_T_ maneuvers by “cutting” inflation time of EIT raw data. Assuming constant gas flow during slow inflation reduction of inflation time, slow inflation V_T_ will decrease proportionally. Moreover, data simulation enabled an increased utilization of existing experimental animal data according to the 3Rs-principle.

Our RVDI-method has not been validated to detect inspiratory overdistension. We did not find relevant amounts of hyperaerated lung tissue, which might be associated with overdistension, using densitometric CT analysis. As addressed previously, the detection of hyperinflation by quantitative CT in pigs is difficult using a threshold of −900 Hounsfield units. In addition, there might be an interaction between threshold levels for density (Hounsfield unit), slice thickness and reconstruction parameters impairing the detection of hyperinflation [41].

## 5. Conclusions

Temporal heterogeneity of regional ventilatory time courses can be quantified by RVDI using EIT during slow inflation maneuvers with a low V_T_, which results in decreased P_EI_. This may reduce the risk for clinical application. In our animal model RVDI-measurements from reduced slow inflation V_T_s are correlated to tidal recruitment. Further research is warranted to reproduce our findings in ARDS patients.

## Figures and Tables

**Figure 1 jcm-08-01161-f001:**
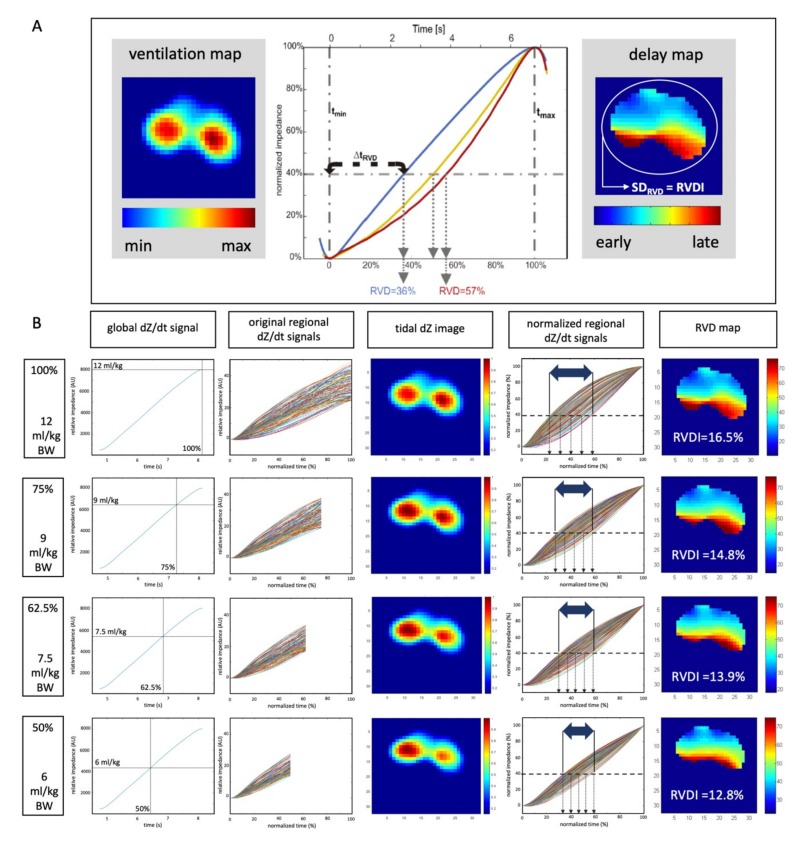
(**A**) Left ventilation map (tidal dZ image) from a slow inflation; middle: Normalized regional impedance/time curves of three exemplary pixels, blue: Early aerated, yellow: Delayed aeration, red: Very late aeration. Regional-Ventilation-Delay time (*Δt*_RVD_) as the difference between the start of inspiration and the time when the respective regional curve *∆Z_i_(t)* reaches a threshold of 40% of the maximal local impedance change. RVD = *Δt*_RVD_*/Δt_max − min;_* right: Delay map: Color coding of pixel-wise regional-ventilation-delay (RVD) indices; Regional ventilation delay inhomogeneity (RVDI): Quantification of temporal heterogeneity (regional ventilation delay inhomogeneity). (**B**) Schematic description; calculation of RVD indices and RVDI from the original 12 mL/kg inflation (analyses from 100% inflation time) and simulation of inflations of reduced inflation volumes of 9, 7.5, and 6 mL/kg (analyses from electrical impedance tomography (EIT) signals truncated after 75%, 62.5%, and 50% inflation time). Images show global dZ/dt curves, original regional dZ/dt curves, tidal dZ images (ventilation map), normalized regional dZ/dt curves for RVD analyses, RVD maps (scaled in % inflation time) and resulting RVDI values.

**Figure 2 jcm-08-01161-f002:**
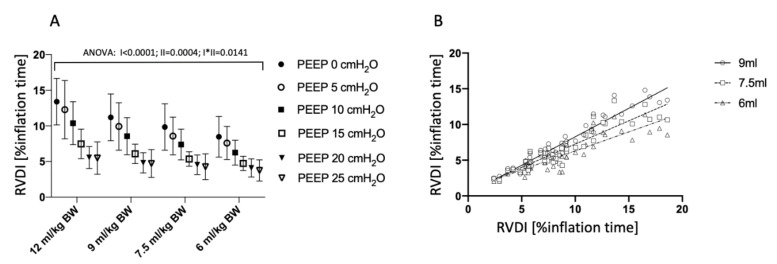
(**A**) Regional ventilation delay inhomogeneity (RVDI) obtained by electrical impedance tomography (EIT) during a slow inflation with tidal volumes of 12, 9, 7.5, and 6 mL/kg at positive end-expiratory pressure (PEEP) levels of 0, 5, 10, 15, 20, and 25 cmH_2_O. Repeated-measures ANOVA: I: factor “PEEP”, II: Factor “slow inflation V_T_”; I*II: Interaction of I and II. (**B**) Linear correlation analysis of RVDI (temporal heterogeneity of regional impedance time curves in % inflation time) obtained from electrical impedance tomography (EIT) during a slow inflation of 12 mL/kg BW (x-axis) and RVDI calculated from simulated 9, 7.5, and 6 mL/kg BW inflations performed at PEEP levels of 0, 5, 10, 15, 20, and 25 cmH_2_O.

**Figure 3 jcm-08-01161-f003:**
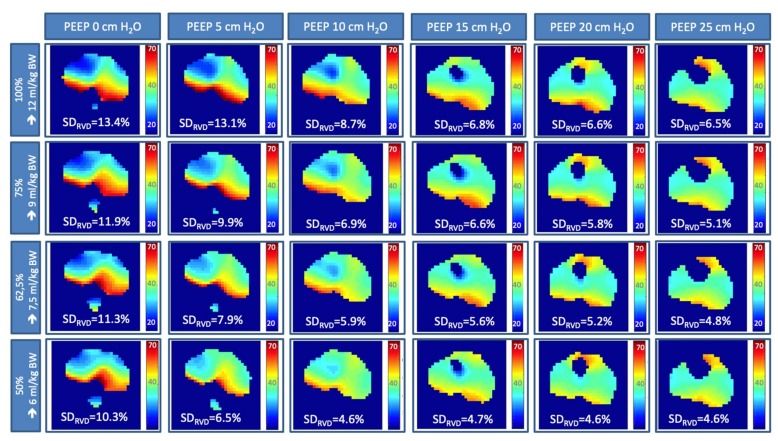
Regional ventilation delay (RVD) analyses, color-coded RVD maps and RVDI values (in % inflation time) obtained by electrical impedance tomography (EIT) during a slow inflation with tidal volumes of 12, 9, 7.5, and 6 mL/kg at PEEP levels of 0, 5, 10, 15, 20, and 25 cmH_2_O.

**Figure 4 jcm-08-01161-f004:**
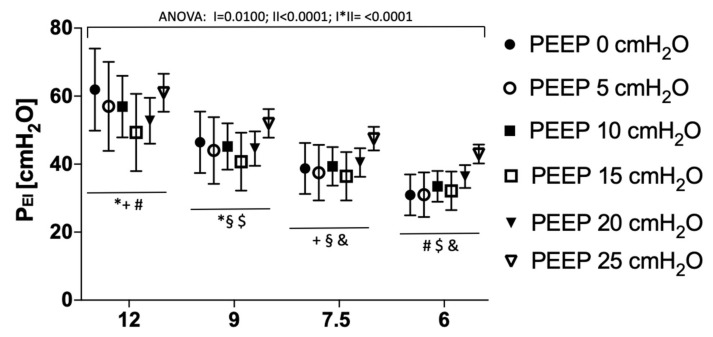
End-inspiratory pressure (P_EI_) resulting from a slow inflation with tidal volumes of 12, 9, 7.5, and 6 mL/kg at PEEP levels of 0, 5, 10, 15, 20, and 25 cmH_2_O. Repeated measures ANOVA: I: Factor “PEEP”, II: Factor “slow inflation V_T_”; I*II: Interaction of I and II; consecutive post hoc tests (Tukey) for differences in P_EI_ between different slow inflation tidal volumes at single PEEP levels: * 12 vs. 9 mL/kg BW, +12 vs. 7.5 mL/kg BW, # 12 vs. 6 mL/kg BW, § 9 vs. 7.5 mL/kg BW, $ 9 vs. 6 mL/kg BW, & 7.5 vs. 6 mL/kg BW, *p* < 0.001, respectively.

**Figure 5 jcm-08-01161-f005:**
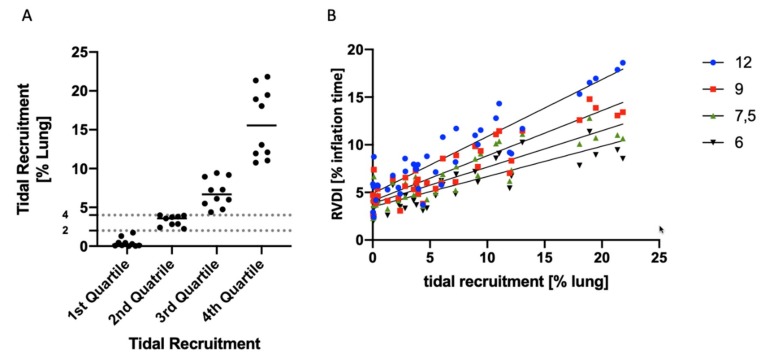
(**A**) Tidal recruitment obtained from computed tomography at PEEP levels of 0, 5, 10, 15, 20, and 25 cmH_2_O in % of total lung volume; given in four quartiles; a threshold of 2% separates the 1st from the 2nd through the 4th quartiles; a threshold of 4% separates the 1st and 2nd from the 3rd and 4th quartiles. (**B**) Linear correlation analysis of tidal recruitment (in % of total lung volume) vs. RVDI (temporal heterogeneity of regional impedance time curves in % inflation time) obtained from electrical impedance tomography (EIT) during a slow inflation of 12, 9, 7.5, and 6 mL/kg.

**Figure 6 jcm-08-01161-f006:**
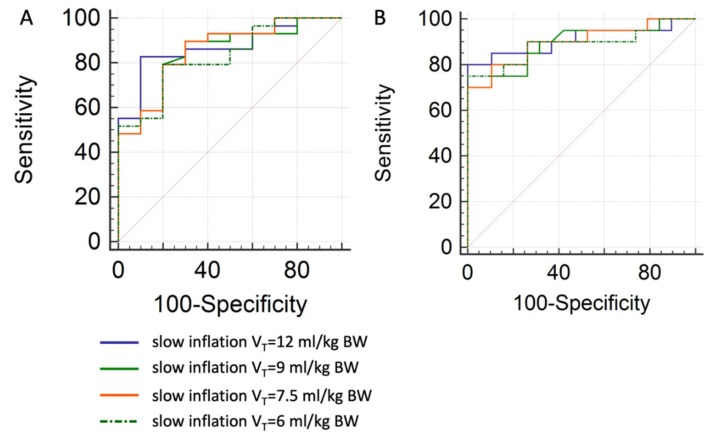
Receiver-Operator-Characteristic (ROC) curve analyses to estimate sensitivity and specificity of certain RVDI values in predicting an increase of tidal recruitment above (**A**) 2% or (**B**) 4% of the total lung volume with decreasing PEEP. Analyses were performed for RVDI values calculated from slow inflation tidal volumes of 12 (blue), 9 (green), 7.5 (yellow), and 6 (doted) mL/kg. Quantitative analyses are given in Table 3.

**Table 1 jcm-08-01161-t001:** Comparison of RVDI values obtained from different slow inflation V_T_ maneuvers using Electrical Impedance Tomography.

Comparison of RVDI Values	Slow Inflation Tidal Volume of 12 mL/kg BW vs.		
	9 mL/kg BW	7.5 mL/kg BW	6 mL/kg BW
inter-individual comparison, all animals (*n* = 7)			
linear correlation, *R*^2^, *p* < 0.001	0.93	0.84	0.76
Bland Altman analysis			
Bias ± SD	−1.5 ± 1.2	−2.4 ± 1.9	−3.3 ± 2.3
95% limits of agreement	−4.0 to 1.0	−6.1 to 1.2	−7.8 to 1.2
intra-individual comparison			
linear correlation, *R*^2^, *p* < 0.001, respectively			
pig 1	0.91	0.79	0.69
pig 2	0.9	0.8	0.73
pig 3	0.97	0.94	0.86
pig 4	0.99	0.99	0.96
pig 5	0.99	0.99	0.97
pig 6	0.85	0.62	0.37
pig 7	0.98	0.94	0.86

Regional ventilation delay inhomogeneity (RVDI) was measured using electrical impedance tomography (EIT) during a slow inflation maneuver to quantify heterogeneity of regional ventilatory time courses.

**Table 2 jcm-08-01161-t002:** Linear correlation of tidal recruitment and RVDI values obtained from different slow inflation V_T_ maneuvers.

Comparison of Tidal Recruitment Versus	RVDI Values Calculated from a Slow Inflation Tidal Volume of			
	12 mL/kg BW	9 mL/kg BW	7.5 mL/kg BW	6 mL/kg BW
inter-individual comparison, all animals (*n* = 7)				
linear correlation, *R*^2^, *p* < 0.001	0.83	0.83	0.76	0.71
intra-individual comparison				
linear correlation, *R*^2^, *p* < 0.001, respectively				
pig 1	0.97	0.91	0.81	0.72
pig 2	0.91	0.95	0.95	0.94
pig 3	0.96	0.94	0.94	0.86
pig 4	0.95	0.95	0.91	0.82
pig 5	0.83	0.77	0.75	0.74
pig 6	0.87	0.62	0.33	0.12
pig 7	0.9	0.93	0.98	0.78

Tidal recruitment was derived from spiral computed tomography. Regional ventilation delay inhomogeneity (RVDI) was measured using electrical impedance tomography (EIT) during a slow inflation maneuver to quantify heterogeneity of regional ventilatory time courses.

**Table 3 jcm-08-01161-t003:** Receiver-Operator-Characteristic curve analyses to predict tidal recruitment from electrical impedance tomography.

ROC Curve Analysis	RVDI Calculated from a Slow Inflation Tidal Volume of			
	12 mL/kg BW	9 mL/kg BW	7.5 mL/kg BW	6 mL/kg BW
tidal recruitment above 2% of lung volume				
AUC	0.869	0.847	0.855	0.828
SE	0.0597	0.0688	0.0680	0.0708
95% CI	0.722 to 0.955	0.695 to 0.942	0.706 to 0.947	0.673 to 0.929
tidal recruitment above 4% of lung volume				
AUC	0.908	0.891	0.897	0.887
SE	0.0533	0.0537	0.0521	0.0583
95% CI	0.771 to 0.977	0.749 to 0.968	0.758 to 0.971	0.744 to 0.966
RVDI-threshold-based prediction of an increase in tidal recruitment above 2 (% lung) during decremental PEEP titration				
RVDI criterion (% inflation time)	>5.36	>4.39	>4.15	>3.34
Sensitivity (%)	90	90	90	90
Specificity (%)	40	50	60	40
RVDI-threshold-based prediction of an increase in tidal recruitment above 4 (% lung) during decremental PEEP titration				
RVDI criterion (% inflation time)	>6.77	>5.66	>4.88	>4.66
Sensitivity (%)	90	90	90	90
Specificity (%)	63	68	74	74

Tidal recruitment was derived from spiral computed tomography. Regional ventilation delay inhomogeneity (RVDI) was measured using electrical impedance tomography (EIT) during a slow inflation maneuver to quantify heterogeneity of regional ventilatory time courses. Receiver-Operator-Characteristic (ROC) curve; AUC: area under the curve; SE: standard error; CI confidence interval.

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
