# Peer review of "A Modified Method to Assess Tidal Recruitment by Electrical Impedance Tomography"

_jcm, 2019, doi:10.3390/jcm8081161_

Round 1
Reviewer 1 Report
Recently it has been discussed that uneven ventilation distribution can cause VILI.
Muders and colleagues conducted a beautiful designed study to safely perform RVDI that provides information about temporal heterogeneity during positive pressure ventilation. They found that tidal volume of 9ml/kg is feasible to measure RVDI.
Results were logically analyzed and conclusion is reasonable.
However, this reviewer does not understand the reason that the specific tidal volumes (6, 7.5, 9) were selected. All of us will agree that tidal volume of 12 ml/kg is hazardous, but we are not sure how safe 9ml/kg is. The golden number of tidal volume is 6 and 6 to 8 ml/kg is acceptable in human beings. Although the authors justified 9ml/kg, safety issue still remains unsolved.
Also, as the authors wrote in Limitation, the simulation to assume data of lower tidal volumes remains to be discussed.
Reviewer 2 Report
It is an extended analysis of a previously published study by the authors. (Tidal recruitment assessed by electrical impedance tomography and computed tomography in a porcine model of lung injury. Crit Care Med 2012; 40:903–911). I think this manuscript still has some merit as it provided some additional information for using RVDI as a surrogate for tidal recruitment during PEEP titration process in an animal model of ARDS. The drawbacks is the number of animals studied is only 7. Number of PEEP dataset for each animal is 6 for each VT protocol (6 to 12ml/Kg BW). The architecture of the manuscript left little to be improved as it is of the same style as published in Crit Care Med. The authors are so comprehensive with their concepts they wrote in this manuscript. The use of RVDI as surrogate about tidal recruitment has limitation mainly becuase of the physiological variability in animal or human ARDS.
As a manuscript from re-analysis of published article, the methdologies and even the viewpoint remained the same except 2 new findings provided. One was slow inflation under reduced VT ( 6 to 9ml/Kg, from truncated original recording) may provide similar information about RVDI as original VT of 12ml/Kg. The other was the thresolds of RVDI for anticipating tidal recruitment above 2 or above 4% lung volume were given. The authors found that RVDI was not satisfactory for anticipating tidal recruitment above 2% but better for predicting tidal recruitment above 4%. As the sensitivity (90%) and specificity ( 63-74%) much improved with RVDI -threshold-based prediction of an increase in tidal recruitment above 4 %. I recommend that the author could say that RVDI could be used for predicting tidal recruitment above 4 % although threhold may be varied with different VT.
I did not agree with author’s viewpoint that tidal recruitment can sufficiently be monitored by EIT-based RVDI-calculation with a slow inflation of 9ml/kg BW only. From the data, it seems that even with a tidal volume of 6ml/Kg, low constant flow inflation could still generate RVDI with similar AUC and sensitivity, specificity to tidal volume of 12ml/Kg. Therefore I suggested emphasizing the use of low flow and low VT( 6-9ml/Kg) may provide information of valuable RVDI for final PEEP titration just as VT of 12ml/Kg. With further reduced VT, it should be less risk for clinical application.
The author mentioned that clinical applicability of strategies aiming at definite RVDI-thresholds may be limited. However, with reduced VT, series RVDI at various PEEP may provide valuable information as shown in the study by Nesler et al for selecting the peronalized PEEP level. This viewpoint is important and already mentioned by the authors.
Reviewer 3 Report
In this re-analysis of experimental study the authors tested the hypothesis using the Regional Ventilation Delay Inhomogeneity (RVDI) to assess the tidal recruitment, derived from slow inflation maneuvers with reduced VT. Correctly the authors reported that the use of high tidal volume (12ml/Kg) limit the clinical applicability when apply high PEEP. Tidal recruitment was quantified at PEEP levels changed stepwise between 0 and 625pxH2O and RVDI was measured by EIT during slow inflation VTs of 12, 9, 7.5, and 6ml/kg BW. Overall the study is interesting and the hypothesis of the study is clear.
However, I have the following comments that the authors should be address:
- First of all, my concern is about the clinical applicability of this method. The authors showed that tidal recruitment can be monitored by EIT-based RVDI-calculation with a slow inflation with Vt 9ml/kg BW. They showed a strong correlation for RVDI values calculated from a slow inflation VT of 9 ml/kg but the correlation is moderate when the Vt during the slow inflation. Probably this method can be able to identify the recruitment but this small changes in terms of Tidal volume (from 12ml/kg to 9ml/Kg) is not able to avoid the overdistenstion. The methods seems not able to identify the overdistension. The authors should discuss this point and add it in the limitation section. Moreover, the authors describe a Vt maneuver that, per se, is a slow-inflation Pressure-volume maneuver. The different – although simulated - tidal volumes produced different pressures during the inflation. Since the aim of the study is to demonstrate that is possible to obtain a reliable RDVI at lower PEI, it is crucial that the authors provide the PEI derived from the different VT. This key information is missing and is fundamental to evaluate the clinical importance of a reduced VT during the maneuver.
- How do you perform the randomization of PEEP level? This randomization is pre-defined?
- In the results section are mentioned the hemodynamic effects of different PEEP-levels. (reported in a previous paper) What are the hemodynamic effects determined by the application of positive airway pressure of 50 cmH2O for 40 seconds before changing PEEP settings? Do you have hemodynamic impairment after this maneuver? How do you manage the hypotension during this maneuver? In how many cases do you stopped the maneuver? Please clarify the choice of this maneuver.
- Please give further explanation of how CT derived tidal recruitment was quantified and which were the ventilatory parameters (especially TV) during the CT scans.
- Page 2, line 45-47. In this context, please cite also more recent papers comparing global and regional lung mechanics ( Scaramuzzo et al. Critical Care 2019 “Heterogeneity of regional inflection points from pressure-volume curves assessed by electrical impedance tomography”)
- Page 3 lines 121-123. Please clarify how do you simulate the slow inflation maneuvers with a VT of 9, 7.5, and 6 ml/kg BW from our 12 ml/kg BW. This point is not clear enough. Furthermore, how do you assume to perform the “analysis of RVD to the first 75%, 62.5%, and 50% of the 12 ml/kg BW maneuvers”. This is a simulation analysis that required a clinical evaluation at bedside. My critical point of view is How this method can be applied at bedside?
- Page 10 lines 255-258. Please re-phrase the main findings of the study related strictly to the results of the study. In my opinion, the main finding of this study is that “ Reduction of the slow inflation VT to 9 ml/kg BW seems to be a good compromise allowing PEI to decrease without relevant loss in clinical information” I suggest to avoid general comments to avoid misunderstanding in the interpretation of the results.
- Page 10 lines 286-289: Please update this statement with new approaches proposed to individualize PEEP titration and analyse the collapse and overdistension. (Please mentioned Spadaro et al Variation of poorly ventilated lung units (silent spaces) measured by electrical impedance tomography to dynamically assess recruitment. Critical Care 2018).
- Figure 3, please fix the y axis (repeated Kg KG)
- Please homogenize acronyms in the manuscript
- Figure 5: please increase the quality of the figure
- Please moderate the conclusion of manuscript and reformulate on the basis of the results.
Limitations: please discuss the fact that tidal recruitment is referred to a tidal ventilation of 6-8 ml kg , since CT scans were acquired using this tidal volume
Round 2
Reviewer 3 Report
Thanks to the authors. The authors addressed to all my requests and the quality of the manuscript was improved. I don't have additional comments.